# Discriminative Robust Transformation Learning

**Jiaji Huang**      **Qiang Qiu**      **Guillermo Sapiro**      **Robert Calderbank**

Department of Electrical Engineering, Duke University
Durham, NC 27708
{`jiaji.huang,qiang.qiu,guillermo.sapiro,robert.calderbank`}@duke.edu

## Abstract

This paper proposes a framework for learning features that are robust to data variation, which is particularly important when only a limited number of training samples are available. The framework makes it possible to tradeoff the discriminative value of learned features against the generalization error of the learning algorithm. Robustness is achieved by encouraging the transform that maps data to features to be a local isometry. This geometric property is shown to improve $(K, \epsilon)$-robustness, thereby providing theoretical justification for reductions in generalization error observed in experiments. The proposed optimization framework is used to train standard learning algorithms such as deep neural networks. Experimental results obtained on benchmark datasets, such as labeled faces in the wild, demonstrate the value of being able to balance discrimination and robustness.

## 1   Introduction

Learning features that are able to discriminate is a classical problem in data analysis. The basic idea is to reduce the variance within a class while increasing it between classes. One way to implement this is by regularizing a certain measure of the variance, while assuming some prior knowledge about the data. For example, Linear Discriminant Analysis (LDA) [4] measures sample covariance and implicitly assumes that each class is Gaussian distributed. The Low Rank Transform (LRT) [10], instead uses nuclear norm to measure the variance and assumes that each class is near a low-rank subspace. A different approach is to regularize the pairwise distances between data points. Examples include the seminal work on metric learning [17] and its extensions [5, 6, 16].

While great attention has been paid to designing objectives to encourage discrimination, less effort has been made in understanding and encouraging robustness to data variation, which is especially important when a limited number of training samples are available. One exception is [19], which promotes robustness by regularizing the traditional metric learning objective using prior knowledge from an auxiliary unlabeled dataset.

In this paper we develop a general framework for balancing discrimination and robustness. Robustness is achieved by encouraging the learned data-to-features transform to be locally an isometry within each class. We theoretically justify this approach using $(K, \epsilon)$-robustness [1, 18] and give an example of the proposed formulation, incorporating it in deep neural networks. Experiments validate the capability to trade-off discrimination against robustness. Our main contributions are the following: 1) prove that locally near isometry leads to robustness; 2) propose a practical framework that allows to robustify a wide class of learned transforms, both linear and nonlinear; 3) provide an explicit realization of the proposed framework, achieving competitive results on difficult face verification tasks.

The paper is organized as follows. Section 2 motivates the proposed study and proposes a general formulation for learning a Discriminative Robust Transform (DRT). Section 3 provides a theoretical justification for the framework by making an explicit connection to robustness. Section 4 gives a

specific example of DRT, denoted as *Euc-DRT*. Section 5 provides experimental validation of Euc-DRT, and section 6 presents conclusions. [1]

## 2 Problem Formulation

Consider an $L$-way classification problem. The training set is denoted by $\mathcal{T} = \{(\mathbf{x}_i, y_i)\}$, where $\mathbf{x}_i \in \mathbb{R}^n$ is the data and $y_i \in \{1, \dots, L\}$ is the class label. We want to learn a feature transform $f_{\boldsymbol{\alpha}}(\cdot)$ such that a datum $\mathbf{x}$ becomes more discriminative when it is transformed to feature $f_{\boldsymbol{\alpha}}(\mathbf{x})$. The transform $f_{\boldsymbol{\alpha}}$ is parametrized by a vector $\boldsymbol{\alpha}$, a framework that includes linear transforms and neural networks where the entries of $\boldsymbol{\alpha}$ are the learned network parameters.

### 2.1 Motivation

The transform $f_{\boldsymbol{\alpha}}$ promotes discriminability by reducing intra-class variance and enlarging inter-class variance. This aim is expressed in the design of objective functions [5, 10] or the structure of the transform $f_{\boldsymbol{\alpha}}$ [7, 11]. However the robustness of the learned transform is an important issue that is often overlooked. When training samples are scarce, statistical learning theory [15] predicts overfitting to the training data. The result of overfitting is that discrimination achieved on test data will be significantly worse than that on training data. Our aim in this paper is the design of robust transforms $f_{\boldsymbol{\alpha}}$ for which the training-to-testing degradation is small [18].

We formally measure robustness of the learned transform $f_{\boldsymbol{\alpha}}$ in terms of $(K, \epsilon)$-robustness [1]. Given a distance metric $\rho$, a learning algorithm is said to be $(K, \epsilon)$-robust if the input data space can be partitioned into $K$ disjoint sets $S_k, k = 1, ..., K$, such that for all training sets $\mathcal{T}$, the learned parameter $\boldsymbol{\alpha}_{\mathcal{T}}$ determines a loss for which the value on pairs of training samples taken from different sets $S_j$ and $S_k$ is very close to the value of any pair of data samples taken from $S_j$ and $S_k$.

$(K, \epsilon)$-robustness is illustrated in Fig. 1, where $S_1$ and $S_2$ are both of diameter $\gamma$ and
$$|e - e'| = |\rho(f_{\boldsymbol{\alpha}}(\mathbf{x}_1), f_{\boldsymbol{\alpha}}(\mathbf{x}_2)) - \rho(f_{\boldsymbol{\alpha}}(\mathbf{x}_1'), f_{\boldsymbol{\alpha}}(\mathbf{x}_2'))|.$$
If the transform $f_{\boldsymbol{\alpha}}$ preserves all distances within $S_1$ and $S_2$, then $|e - e'|$ cannot deviate much from $|d - d'| \le 2\gamma$.

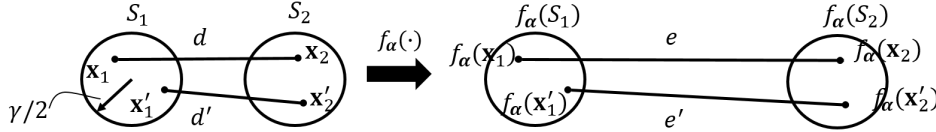

Figure 1: $(K, \epsilon)$-robustness: Here $d = \rho(\mathbf{x}_1, \mathbf{x}_2)$, $d' = \rho(\mathbf{x}_1', \mathbf{x}_2')$, $e = \rho(f_{\boldsymbol{\alpha}}(\mathbf{x}_1), f_{\boldsymbol{\alpha}}(\mathbf{x}_2))$, and $e' = \rho(f_{\boldsymbol{\alpha}}(\mathbf{x}_1'), f_{\boldsymbol{\alpha}}(\mathbf{x}_2'))$. The difference $|e - e'|$ cannot deviate too much from $|d - d'|$.

### 2.2 Formulation and Discussion

Motivated by the above reasoning, we now present our proposed framework. First we define a pair label $\ell_{i,j} \triangleq \begin{cases} 1 & \text{if } y_i = y_j \\ -1 & \text{otherwise} \end{cases}$. Given a metric $\rho$, we use the following hinge loss to encourage high inter-class distance and small intra-class distance.

$$\frac{1}{|\mathcal{P}|} \sum_{i,j \in \mathcal{P}} \max \left\{ 0, \ell_{i,j} \left[ \rho \left( f_{\boldsymbol{\alpha}}(\mathbf{x}_i), f_{\boldsymbol{\alpha}}(\mathbf{x}_j) \right) - t(\ell_{i,j}) \right] \right\}, \tag{1}$$

Here $\mathcal{P} = \{(i, j | i \ne j)\}$ is the set of all data pairs. $t(\ell_{i,j}) \ge 0$ is a function of $\ell_{i,j}$ and $t(1) < t(-1)$. Similar to metric learning [17], this loss function connects pairwise distance to discrimination. However traditional metric learning typically assumes squared Euclidean distance and here the metric $\rho$ can be arbitrary.

For robustness, as discussed above, we may want $f_{\boldsymbol{\alpha}}(\cdot)$ to be distance-preserving within each small local region. In particular, we define the set of all local neighborhoods as
$$\mathcal{NB} \triangleq \{(i, j) | \ell_{i,j} = 1, \rho(\mathbf{x}_i, \mathbf{x}_j) \le \gamma\}.$$

Therefore, we minimize the following objective function

$$\frac{1}{|\mathcal{NB}|} \sum_{(i,j)\in\mathcal{NB}} |\rho(f_{\boldsymbol{\alpha}}(\mathbf{x}_i), f_{\boldsymbol{\alpha}}(\mathbf{x}_j)) - \rho(\mathbf{x}_i, \mathbf{x}_j)|. \tag{2}$$

Note that we do not need to have the same metric in both the input and the feature space, they do not even have in general the same dimension. With a slight abuse of notation we use the same symbol to denote both metrics.

To achieve discrimination and robustness simultaneously, we formulate the objective function as a weighted linear combination of the two extreme cases in (1) and (2)

$$\frac{\lambda}{|\mathcal{P}|} \sum_{i,j\in\mathcal{P}} \max\left\{0, \ell_{i,j}\left[\rho\left(f_{\boldsymbol{\alpha}}(\mathbf{x}_i), f_{\boldsymbol{\alpha}}(\mathbf{x}_j)\right) - t(\ell_{i,j})\right]\right\} + \frac{1-\lambda}{|\mathcal{NB}|} \sum_{(i,j)\in\mathcal{NB}} |\rho(f_{\boldsymbol{\alpha}}(\mathbf{x}_i), f_{\boldsymbol{\alpha}}(\mathbf{x}_j)) - \rho(\mathbf{x}_i, \mathbf{x}_j)|$$
$$\tag{3}$$

where $\lambda \in [0, 1]$. The formulation (3) balances discrimination and robustness. When $\lambda = 1$ it seeks discrimination, and as $\lambda$ decreases it starts to encourage robustness. We shall refer to a transform that is learned by solving (3) as a Discriminative Robust Transform (DRT). The DRT framework provides opportunity to select both the distance measure and the transform family.

## 3 Theoretical Analysis

In this section, we provide a theoretical explanation for robustness. In particular, we show that if the solution to (1) yields a transform $f_{\boldsymbol{\alpha}}$ that is locally a near isometry, then $f_{\boldsymbol{\alpha}}$ is robust.

### 3.1 Theoretical Framework

Let $\mathcal{X}$ denote the original data, let $\mathcal{Y} = \{1, ..., L\}$ denote the set of class labels, and let $\mathcal{Z} = \mathcal{X} \times \mathcal{Y}$. The training samples are pairs $\mathbf{z}_i = (\mathbf{x}_i, y_i), i = 1, \ldots, n$ drawn from some unknown distribution $\mathcal{D}$ defined on $\mathcal{Z}$. The indicator function is defined as $\ell_{i,j} = 1$ if $y_i = y_j$ and $-1$ otherwise. Let $f_{\boldsymbol{\alpha}}$ be a transform that maps a low-level feature $\mathbf{x}$ to a more discriminative feature $f_{\boldsymbol{\alpha}}(\mathbf{x})$, and let $\mathcal{F}$ denote the space of transformed features.

For simplicity we consider an arbitrary metric $\rho$ defined on both $\mathcal{X}$ and $\mathcal{F}$ (the general case of different metrics is a straightforward extension), and a loss function $g(\rho(f_{\boldsymbol{\alpha}}(\mathbf{x}_i), f_{\boldsymbol{\alpha}}(\mathbf{x}_j)), \ell_{i,j})$ that encourages $\rho(f_{\boldsymbol{\alpha}}(\mathbf{x}_i), f_{\boldsymbol{\alpha}}(\mathbf{x}_j))$ to be small (big) if $\ell_{i,j} = 1$ $(-1)$. We shall require the Lipschtiz constant of $g(\cdot, 1)$ and $g(\cdot, -1)$ to be upper bounded by $A > 0$. Note that the loss function in Eq. (1) has a Lipschtiz constant of 1. We abbreviate

$$g(\rho(f_{\boldsymbol{\alpha}}(\mathbf{x}_i), f_{\boldsymbol{\alpha}}(\mathbf{x}_j)), \ell_{i,j}) \triangleq h_{\boldsymbol{\alpha}}(\mathbf{z}_i, \mathbf{z}_j).$$

The empirical loss on the training set is a function of $\boldsymbol{\alpha}$ given by

$$R_{emp}(\boldsymbol{\alpha}) \triangleq \frac{2}{n(n-1)} \sum_{\substack{i,j=1 \\ i\neq j}}^{n} h_{\boldsymbol{\alpha}}(\mathbf{z}_i, \mathbf{z}_j), \tag{4}$$

and the expected loss on the test data is given by

$$R(\boldsymbol{\alpha}) \triangleq \mathbb{E}_{\mathbf{z}_1', \mathbf{z}_2' \sim \mathcal{D}} \left[ h_{\boldsymbol{\alpha}}(\mathbf{z}_1', \mathbf{z}_2') \right]. \tag{5}$$

The algorithm operates on pairs of training samples and finds parameters

$$\boldsymbol{\alpha}_{\mathcal{T}} \triangleq \arg\min_{\boldsymbol{\alpha}} R_{emp}(\boldsymbol{\alpha}), \tag{6}$$

that minimize the empirical loss on the training set $\mathcal{T}$. The difference $R_{emp} - R$ between expected loss on the test data and empirical loss on the training data is the generalization error of the algorithm.

### 3.2 $(K, \epsilon)$-robustness and Covering Number

We work with the following definition of $(K, \epsilon)$-robustness [1].
**Definition 1.** *A learning algorithm is $(K, \epsilon)$-robust if $\mathcal{Z} = \mathcal{X} \times \mathcal{Y}$ can be partitioned into $K$ disjoint sets $\mathcal{Z}_k, k = 1, \ldots, K$ such that for all training sets $\mathcal{T} \in \mathcal{Z}^n$, the learned parameter $\boldsymbol{\alpha}_{\mathcal{T}}$ determines a loss function where the value on pairs of training samples taken from sets $\mathcal{Z}_p$ and $\mathcal{Z}_q$ is "very close" to the value of any pair of data samples taken from $\mathcal{Z}_p$ and $\mathcal{Z}_q$. Formally, assume $\mathbf{z}_i, \mathbf{z}_j \in \mathcal{T}$, with $\mathbf{z}_i \in \mathcal{Z}_p$ and $\mathbf{z}_j \in \mathcal{Z}_q$, if $\mathbf{z}_i' \in \mathcal{Z}_p$ and $\mathbf{z}_j' \in \mathcal{Z}_q$, then*

$$\left| h_{\boldsymbol{\alpha}_{\mathcal{T}}}(\mathbf{z}_i, \mathbf{z}_j) - h_{\boldsymbol{\alpha}_{\mathcal{T}}}(\mathbf{z}_i', \mathbf{z}_j') \right| \leq \epsilon.$$

**Remark 1.** *$(K, \epsilon)$-robustness means that the loss incurred by a testing pair $(\mathbf{z}'_i, \mathbf{z}'_j)$ in $\mathcal{Z}_p \times \mathcal{Z}_q$ is very close to the loss incurred by any training pair $(\mathbf{z}_i, \mathbf{z}_j)$ in $\mathcal{Z}_p \times \mathcal{Z}_q$. It is shown in [1] that the generalization error of $(K, \epsilon)$-robust algorithms is bounded as*

$$R(\boldsymbol{\alpha}_{\mathcal{T}}) - R_{emp}(\boldsymbol{\alpha}_{\mathcal{T}}) \leq \epsilon + O\left(\sqrt{\frac{K}{n}}\right). \tag{7}$$

*Therefore the smaller $\epsilon$, the smaller is the generalization error, and the more robust is the learning algorithm.*

Given a metric space, the covering number specifies how many balls of a given radius are needed to cover the space. The more complex the metric space, the more balls are needed to cover it. Covering number is formally defined as follows.

**Definition 2** (Covering number). *Given a metric space $(\mathcal{S}, \rho)$, we say that a subset $\hat{\mathcal{S}}$ of $\mathcal{S}$ is a $\gamma$-cover of $\mathcal{S}$, if for every element $\mathbf{s} \in \mathcal{S}$, there exists $\hat{\mathbf{s}} \in \hat{\mathcal{S}}$ such that $\rho(\mathbf{s}, \hat{\mathbf{s}}) \leq \gamma$. The $\gamma$-covering number of $\mathcal{S}$ is*

$$\mathcal{N}_\gamma(\mathcal{S}, \rho) = \min\{|\hat{\mathcal{S}}| : \hat{\mathcal{S}} \text{ is a } \gamma\text{-cover of } \mathcal{S}\}.$$

**Remark 2.** *The covering number is a measure of the geometric complexity of $(\mathcal{S}, \rho)$. A set $S$ with covering number $\mathcal{N}_{\gamma/2}(\mathcal{S}, \rho)$ can be partitioned into $\mathcal{N}_{\gamma/2}(\mathcal{S}, \rho)$ disjoint subsets, such that any two points within the same subset are separated by no more than $\gamma$.*

**Lemma 1.** *The metric space $\mathcal{Z} = \mathcal{X} \times \mathcal{Y}$ can be partitioned into $L\mathcal{N}_{\gamma/2}(\mathcal{X}, \rho)$ subsets, denoted as $\mathcal{Z}_1, \ldots, \mathcal{Z}_{L\mathcal{N}_{\gamma/2}(\mathcal{X}, \rho)}$, such that any two points $\mathbf{z}_1 \triangleq (\mathbf{x}_1, y_1), \mathbf{z}_2 \triangleq (\mathbf{x}_2, y_2)$ in the same subset satisfy $y_1 = y_2$ and $\rho(\mathbf{x}_1, \mathbf{x}_2) \leq \gamma$.*

*Proof.* Assuming the metric space $(\mathcal{X}, \rho)$ is compact, we can partition $\mathcal{X}$ into $\mathcal{N}_{\gamma/2}(\mathcal{X}, \rho)$ subsets, each with diameter at most $\gamma$. Since $\mathcal{Y}$ is a finite set of size $L$, we can partition $\mathcal{Z} = \mathcal{X} \times \mathcal{Y}$ into $L\mathcal{N}_{\gamma/2}(\mathcal{X}, \rho)$ subsets with the property that two samples $(\mathbf{x}_1, y_1), (\mathbf{x}_2, y_2)$ in the same subset satisfy $y_1 = y_2$ and $\rho(\mathbf{x}_1, \mathbf{x}_2) \leq \gamma$. $\square$

It follows from Lemma 1 that we may partition $\mathcal{X}$ into subsets $\mathcal{X}_1, \ldots, \mathcal{X}_{L\mathcal{N}_{\gamma/2}(\mathcal{X}, \rho)}$, such that pairs of points $\mathbf{x}_1, \mathbf{x}_2$ from the same subset have the same label and satisfy $\rho(\mathbf{x}_i, \mathbf{x}_j) \leq \gamma$. Before we connect local geometry to robustness we need one more definition. We say that a learned transform $f_{\boldsymbol{\alpha}}$ is a $\delta$-isometry if the metric is distorted by at most $\delta$:

**Definition 3** ($\delta$-isometry). *Let $\mathcal{A}, \mathcal{B}$ be metric spaces with metrics $\rho_{\mathcal{A}}$ and $\rho_{\mathcal{B}}$. A map $f: \mathcal{A} \mapsto \mathcal{B}$ is a $\delta$-isometry if for any $\mathbf{a}_1, \mathbf{a}_2 \in \mathcal{A}$, $|\rho_{\mathcal{A}}(f(\mathbf{a}_1), f(\mathbf{a}_2)) - \rho_{\mathcal{B}}(\mathbf{a}_1, \mathbf{a}_2)| \leq \delta$.*

**Theorem 1.** *Let $f_{\boldsymbol{\alpha}}$ be a transform derived via Eq. (6) and let $\mathcal{X}_1, \ldots, \mathcal{X}_{L\mathcal{N}_{\gamma/2}(\mathcal{X}, \rho)}$ be a cover of $\mathcal{X}$ as described above. If $f_{\boldsymbol{\alpha}}$ is a $\delta$-isometry, then it is $(L\mathcal{N}_{\gamma/2}(\mathcal{X}, \rho), 2A(\gamma + \delta))$-robust.*

*Proof sketch.* Consider training samples $\mathbf{z}_i, \mathbf{z}_j$ and testing samples $\mathbf{z}'_i, \mathbf{z}'_j$ such that $\mathbf{z}_i, \mathbf{z}'_i \in \mathcal{Z}_p$ and $\mathbf{z}_j, \mathbf{z}'_j \in \mathcal{Z}_q$ for some $p, q \in \{1, \ldots, L\mathcal{N}_{\gamma/2}(\mathcal{X}, \rho)\}$. Then by Lemma 1,

$$\rho(\mathbf{x}_i, \mathbf{x}'_i) \leq \gamma \text{ and } \rho(\mathbf{x}_j, \mathbf{x}'_j) \leq \gamma, \qquad y_i = y'_i \text{ and } y_j = y'_j,$$

and $\mathbf{x}_i, \mathbf{x}'_i \in \mathcal{X}_p$ and $\mathbf{x}_j, \mathbf{x}'_j \in \mathcal{X}_q$. By definition of $\delta$-isometry,

$$|\rho(f_{\boldsymbol{\alpha}_{\mathcal{T}}}(\mathbf{x}_i), f_{\boldsymbol{\alpha}_{\mathcal{T}}}(\mathbf{x}'_i)) - \rho(\mathbf{x}_i, \mathbf{x}'_i)| \leq \delta \text{ and } |\rho(f_{\boldsymbol{\alpha}_{\mathcal{T}}}(\mathbf{x}_j), f_{\boldsymbol{\alpha}_{\mathcal{T}}}(\mathbf{x}'_j)) - \rho(\mathbf{x}_j, \mathbf{x}'_j)| \leq \delta.$$

Rearranging the terms gives

$$\rho(f_{\boldsymbol{\alpha}_{\mathcal{T}}}(\mathbf{x}_i), f_{\boldsymbol{\alpha}_{\mathcal{T}}}(\mathbf{x}'_i)) \leq \rho(\mathbf{x}_i, \mathbf{x}'_i) + \delta \leq \gamma + \delta \text{ and } \rho(f_{\boldsymbol{\alpha}_{\mathcal{T}}}(\mathbf{x}_j), f_{\boldsymbol{\alpha}_{\mathcal{T}}}(\mathbf{x}'_j)) \leq \rho(\mathbf{x}_j, \mathbf{x}'_j) + \delta \leq \gamma + \delta.$$

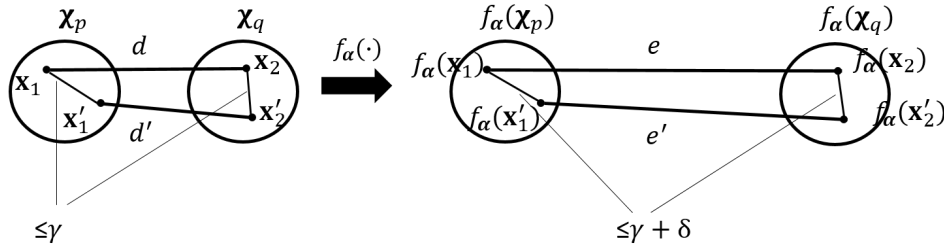

Figure 2: Proof without words.

In order to bound the generalization error, we need to bound the difference between $\rho(f_{\boldsymbol{\alpha}_{\mathcal{T}}}(\mathbf{x}_i), f_{\boldsymbol{\alpha}_{\mathcal{T}}}(\mathbf{x}_j))$ and $\rho(f_{\boldsymbol{\alpha}_{\mathcal{T}}}(\mathbf{x}'_i), f_{\boldsymbol{\alpha}_{\mathcal{T}}}(\mathbf{x}'_j))$. The details can be found in [9]; here we appeal to the proof schematic in Fig. 2. We need to bound $|e - e'|$ and it cannot exceed twice the diameter of a local region in the transformed domain. □

Robustness of the learning algorithm depends on the granularity of the cover and the degree to which the learned transform $f_{\boldsymbol{\alpha}}$ distorts distances between pairs of points in the same covering subset. The subsets in the cover constitute regions where the local geometry makes it possible to bound generalization error. It now follows from [1] that the generalization error satisfies $R(\boldsymbol{\alpha}_{\mathcal{T}}) - R_{emp}(\boldsymbol{\alpha}_{\mathcal{T}}) \leq 2A(\gamma + \delta) + O\left(\sqrt{\frac{K}{n}}\right)$. The DRT proposed here is a particular example of a local isometry, and Theorem 1 explains why the generalization error is smaller than that of pure metric learning.

The transform described in [9] partitions the metric space $\mathcal{X}$ into exactly $L$ subsets, one for each class. The experiments reported in Section 5 demonstrate that the performance improvements derived from working with a finer partition can be worth the cost of learning finer grained local regions.

## 4 An Illustrative Realization of DRT

Having justified robustness, we now provide a realization of the proposed general DRT where the metric $\rho$ is Euclidean distance. We use Gaussian random variables to initialize $\boldsymbol{\alpha}$, then, on the randomly transformed data, we set $t(1)$ ($t(-1)$) to be the average intra-class (inter-class) pairwise distance. In all our experiments, the solution satisfied the condition $t(1) < t(-1)$ required in Eq. (1). We calculate the diameter $\gamma$ of the local regions $\mathcal{NB}$ indirectly, using the $\kappa$-nearest neighbors of each training sample to define a local neighborhood. We leave the question of how best to initialize the indicator $t$ and the diameter $\gamma$ for future research.

We denote this particular example as Euc-DRT and use gradient descent to solve for $\boldsymbol{\alpha}$. Denoting the objective by $J$, we define $\mathbf{y}_i \triangleq f_{\boldsymbol{\alpha}}(\mathbf{x}_i)$, $\delta_{i,j} \triangleq f_{\boldsymbol{\alpha}}(\mathbf{x}_i) - f_{\boldsymbol{\alpha}}(\mathbf{x}_j)$, and $\rho_{i,j}^0 \triangleq \|\mathbf{x}_i - \mathbf{x}_j\|$. Then

$$\frac{\partial J}{\partial \mathbf{y}_i} = \sum_{\substack{(i,j)\in\mathcal{P} \\ \ell_{i,j}(\|\delta_{i,j}\| - t(\ell_{i,j})) > 0}} \frac{\lambda}{|\mathcal{P}|} \cdot \ell_{i,j} \cdot \frac{\delta_{i,j}}{\|\delta_{i,j}\|} + \sum_{(i,j)\in\mathcal{NB}} \frac{1-\lambda}{|\mathcal{NB}|} \cdot \operatorname{sgn}(\|\delta_{i,j}\| - \rho_{i,j}^0) \cdot \frac{\delta_{i,j}}{\|\delta_{i,j}\|}. \quad (8)$$

In general, $f_{\boldsymbol{\alpha}}$ defines a $D$-layer neural network (when $D = 1$ it defines a linear transform). Let $\boldsymbol{\alpha}^{(d)}$ be the linear weights at the $d$-th layer, and let $\mathbf{x}^{(d)}$ be the output of the $d$-th layer, so that $\mathbf{y}_i = \mathbf{x}_i^{(D)}$. Then the gradients are computed as,

$$\frac{\partial J}{\partial \boldsymbol{\alpha}^{(D)}} = \sum_i \frac{\partial J}{\partial \mathbf{y}_i} \cdot \frac{\partial \mathbf{y}_i}{\partial \boldsymbol{\alpha}^{(D)}}, \text{ and } \frac{\partial J}{\partial \boldsymbol{\alpha}^{(d)}} = \sum_i \frac{\partial J}{\partial \mathbf{x}_i^{(d+1)}} \cdot \frac{\partial \mathbf{x}_i^{(d+1)}}{\partial \mathbf{x}_i^{(d)}} \cdot \frac{\partial \mathbf{x}_i^{(d)}}{\partial \boldsymbol{\alpha}^{(d)}} \text{ for } 1 \leq d \leq D-1. \quad (9)$$

Algorithm 1 provides a summary, and we note that the extension to stochastic training using minibatches is straightforward.

## 5 Experimental Results

In this section we report on experiments that confirm robustness of Euc-DRT. Recall that empirical loss is given by Eq. (4) where $\boldsymbol{\alpha}$ is learned as $\boldsymbol{\alpha}_{\mathcal{T}}$ from the training set $\mathcal{T}$, and $|\mathcal{T}| = N$. The generalization error is $R - R_{emp}$ where the expected loss $R$ is estimated using a large test set.

### 5.1 Toy Example

This illustrative example is motivated by the discussion in Section 2.1. We first generate a 2D dataset consisting of two noisy half-moons, then use a random $100 \times 2$ matrix to embed the data in a 100-dimensional space. We learn a linear transform $f_{\boldsymbol{\alpha}}$ that maps the 100 dimensional data to 2 dimensional features, and we use $\kappa = 5$ nearest neighbors to construct the set $\mathcal{NB}$. We consider $\lambda = 1, 0.5, 0.25$, representing the most discriminative, balanced, and more robust scenarios.

When $\lambda = 1$ the transformed training samples are rather discriminative (Fig. 3a), but when the transform is applied to testing data, the two classes are more mixed (Fig. 3d). When $\lambda = 0.5$, the

---

**Algorithm 1** Gradient descent solver for Euc-DRT

---

**Input:** $\lambda \in [0,1]$, training pairs $\{(\mathbf{x}_i, \mathbf{x}_j, \ell_{i,j})\}$, a pre-defined $D$-layer network ($D = 1$ as linear transform), stepsize $\eta$, neighborhood size $\kappa$.

**Output:** $\boldsymbol{\alpha}$

1:  Randomly initialize $\boldsymbol{\alpha}$, compute $\mathbf{y}_i = f_{\boldsymbol{\alpha}}(\mathbf{x}_i)$.
2:  On the $\mathbf{y}_i$, compute the average intra and inter-class pairwise distances, assign to $t(1), t(-1)$
3:  For each training datum, find its $\kappa$ nearest neighbor and define the set $\mathcal{NB}$.
4:  **while** stable objective not achieved **do**
5:      Compute $\mathbf{y}_i = f_{\boldsymbol{\alpha}}(\mathbf{x}_i)$ by a forward pass.
6:      Compute objective $J$.
7:      Compute $\frac{\partial J}{\partial \mathbf{y}_i}$ as Eq. (8).
8:      **for** $l = D$ down to 1 **do**
9:          Compute $\frac{\partial J}{\partial \boldsymbol{\alpha}^{(d)}}$ as Eq. (9).
10:         $\boldsymbol{\alpha}^{(d)} \leftarrow \boldsymbol{\alpha}^{(d)} - \eta \frac{\partial J}{\partial \boldsymbol{\alpha}^{(d)}}$.
11:     **end for**
12: **end while**

---

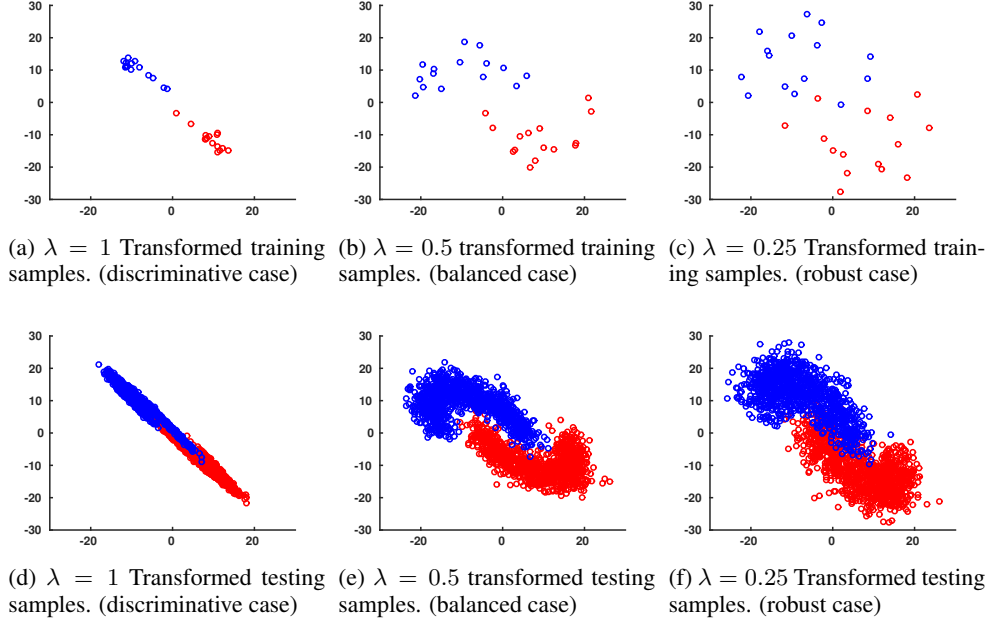

(a) $\lambda = 1$ Transformed training samples. (discriminative case)

(b) $\lambda = 0.5$ transformed training samples. (balanced case)

(c) $\lambda = 0.25$ Transformed training samples. (robust case)

(d) $\lambda = 1$ Transformed testing samples. (discriminative case)

(e) $\lambda = 0.5$ transformed testing samples. (balanced case)

(f) $\lambda = 0.25$ Transformed testing samples. (robust case)

Figure 3: Original and transformed training/testing samples embedded in 2-dimensional space with different colors representing different classes.

transformed training data are more dispersed within each class (Fig. 3b), hence less easily separated than when $\lambda = 1$. However Fig. 3e shows that it is easier to separate the two classes on the test data. When $\lambda = 0.25$, robustness is preferred to discriminative power as shown in Figs. 3c and 3f.

Tab. 1 quantifies empirical loss $R_{emp}$, generalization error, and classification performance (by 1-nn) for $\lambda = 1, 0.5$ and $0.25$. As $\lambda$ decreases, $R_{emp}$ increases, indicating loss of discrimination on the training set. However, generalization error decreases, implying more robustness. We conclude that by varying $\lambda$, we can balance discrimination and robustness.

## 5.2 MNIST Classfication Using a Very Small Training Set

The transform $f_{\boldsymbol{\alpha}}$ learned in the previous section was linear, and we now apply a more sophisticated convolutional neural network to the MNIST dataset. The network structure is similar to LeNet, and is

Table 1: Varying $\lambda$ on a toy dataset.

| $\lambda$ | 1 | 0.5 | 0.25 |
|---|---|---|---|
| $R_{emp}$ | 1.5983 | 1.6025 | 1.9439 |
| generalization error | 10.5855 | 9.5071 | 8.8040 |
| 1-nn accuracy (original data 93.35%) | 92.20% | **98.30%** | 91.55% |

Table 2: Classification error on MNIST.

| Training/class | 30 | 50 | 70 | 100 |
|---|---|---|---|---|
| original pixels | 81.91% | 86.18% | 86.86% | 88.49% |
| LeNet | 87.51% | 89.89% | 91.24% | 92.75% |
| DML | 92.32% | 94.45% | 95.67% | 96.19% |
| Euc-DRT | **94.14%** | **95.20%** | **96.05%** | **96.21%** |

Table 3: Implementation details of the neural network for MNIST classification.

| name | parameters |
|---|---|
| conv1 | size: $5 \times 5 \times 1 \times 20$ stride: 1, pad: 0 |
| pool1 | size: $2 \times 2$ |
| conv2 | size: $5 \times 5 \times 20 \times 50$ stride: 1, pad: 0 |
| pool2 | size: $2 \times 2$ |
| conv3 | size: $4 \times 4 \times 50 \times 128$ stride: 1, pad: 0 |

made up of alternating convolutional layers and pooling layers, with parameters detailed in Table 3. We map the original 784-dimensional pixel values (28x28 image) to 128-dimensional features.

While state-of-art results often use the full training set (6,000 training samples per class), here we are interested in small training sets. We use only 30 training samples per class, and we use $\kappa = 7$ nearest neighbors to define local regions in Euc-DRT. We vary $\lambda$ and study empirical error, generalization error, and classification accuracy (1-nn). We observe in Fig. 4 that when $\lambda$ decreases, the empirical error also decreases, but that the generalization error actually increases. By balancing between these two factors, a peak classification accuracy is achieved at $\lambda = 0.25$. Next, we use 30, 50, 70, 100

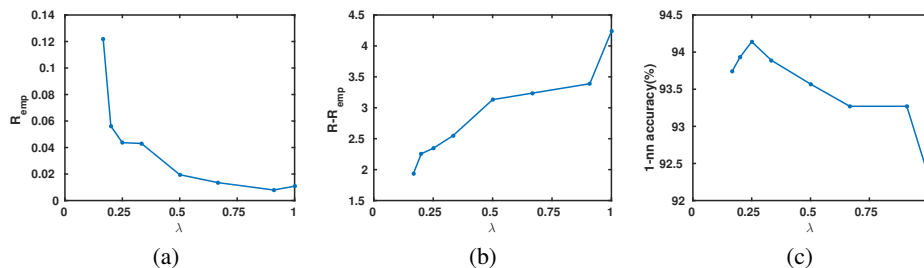

(a)                    (b)                    (c)

Figure 4: MNIST test: with only 30 training samples per class. We vary $\lambda$ and assess (a) $R_{emp}$; (b) generalization error; and (c) 1-nn classification accuracy. Peak accuracy is achieved at $\lambda = 0.25$.

training samples per class and compare the performance of Euc-DRT with LeNet and Deep Metric Learning (DML) [7]. DML minimizes a hinge loss on the squared Euclidean distances. It shares the same spirit with our Euc-DRT using $\lambda = 1$. All methods use the same network structure, Tab. 3, to map to the features. For classification, LeNet uses a linear softmax classifier on top of the "conv3" layer and minimizes the standard cross-entropy loss during training. DML and Euc-DRT both use a 1-nn classifier on the learned features. Classification accuracies are reported in Tab. 2. In Tab. 2, we see that all the learned features improve upon the original ones. DML is very discriminative and achieves higher accuracy than LeNet. However, when the training set is very small, robustness becomes more important and Euc-DRT significantly outperforms DML.

## 5.3 Face Verification on LFW

We now present face verification on the more challenging Labeled Faces in the Wild (LFW) benchmark, where our experiments will show that there is an advantage to balancing discriminability and robustness. Our goal is not to reproduce the success of deep learning in face verification [7, 14], but to stress the importance of robust training and to compare the proposed Euc-DRT objective with popular alternatives. Note also that it is difficult to compare with deep learning methods when training sets are proprietary [12–14].

We adopt the experimental framework used in [2], and train a deep network on the WDRef dataset, where each face is described using a high dimensional LBP feature [3] (available at [2]) that is reduced to a 5000-dimensional feature using PCA. The WDRef dataset is significantly smaller than the proprietary datasets typical of deep learning, such as the 4.4 million labeled faces from 4030 individuals in [14], or the 202,599 labeled faces from 10,177 individuals in [12]. It contains 2,995 subjects with about 20 samples per subject.

We compare the Euc-DRT objective with DeepFace (DF) [14] and Deep Metric Learning (DML) [7], two state-of-the-art deep learning objectives. For a fair comparison, we employ the same network structure and train on the same input data. DeepFace feeds the output of the last network layer to an $L$-way soft-max to generate a probability distribution over $L$ classes, then minimizes a cross entropy loss. The Euc-DRT feature $f_\alpha$ is implemented as a two-layer fully connected network with *tanh* as the squash function. Weight decay (conventional Frobenius norm regularization) is employed in both DF and DML, and results are only reported for the best weight decay factor. After a network is trained on WDRef, it is tested on the LFW benchmark. Verification simply consists of comparing the cosine distance between a given pair of faces to a threshold.

Fig. 5 displays ROC curves and Table 4 reports area under the ROC curve (AUC) and verification accuracy. High-Dim LBP refers to verification using the initial LBP features. DeepFace (DF) optimizes for a classification objective by minimizing a softmax loss, and it successfully separates samples from different classes. However the constraint that assigns similar representations to the same class is weak, and this is reflected in the true positive rate displayed in Fig. 5. In Deep Metric Learning (DML) this same constraint is strong, but robustness is a concern when the training set is small. The proposed Euc-DRT improves upon both DF and DML by balancing discriminability and robustness. It is less conservative than DF for better discriminability, and more responsive to local geometry than DML for smaller generalization error. Face verification accuracy for Euc-DRT was obtained by varying the regularization parameter $\lambda$ between 0.4 and 1 (as shown in Fig 6), then reporting the peak accuracy observed at $\lambda = 0.9$.

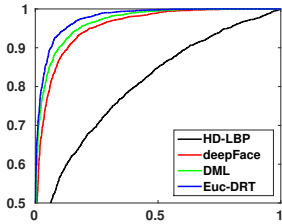

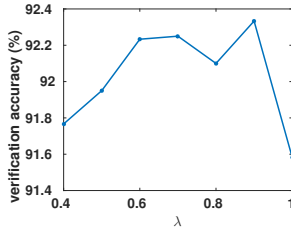

Figure 5: Comparison of ROCs for all methods

Figure 6: Verification accuracy of Euc-DRT as $\lambda$ varies

Table 4: Verification accuracy and AUCs on LFW

| Method | Accuracy (%) | AUC ($\times 10^{-2}$) |
|---|---|---|
| HD-LBP | 74.73 | 82.22±1.00 |
| deepFace | 88.72 | 95.50± 0.29 |
| DML | 90.28 | 96.74±0.33 |
| Euc-DRT | **92.33** | **97.77± 0.25** |

## 6  Conclusion

We have proposed an optimization framework within which it is possible to tradeoff the discriminative value of learned features with robustness of the learning algorithm. Improvements to generalization error predicted by theory are observed in experiments on benchmark datasets. Future work will investigate how to initialize and tune the optimization, also how the Euc-DRT algorithm compares with other methods that reduce generalization error.

## 7  Acknowledgement

The work of Huang and Calderbank was supported by AFOSR under FA 9550-13-1-0076 and by NGA under HM017713-1-0006. The work of Qiu and Sapiro is partially supported by NSF and DoD.

## Footnotes

[1]A note on the notations: matrices (vectors) are denoted in upper (lower) case bold letters. Scalars are denoted in plain letters.

[2]http://home.ustc.edu.cn/chendong/

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
