[Reviews · NeurIPS 2015]

Submitted by Assigned_Reviewer_1

This paper presents a method to learn a transformation using a loss function in which it is possible to control the trade-off between robustness and class discrimination. The loss function is a linear combination of a discriminative element, which tries to maximise the between-class separation and a robustness element, which tries to preserve isometry between source feature space and target (classification) mapping. The method is theoretically justified based on the metric learning method of [1], thought the focus here is on learning a mapping function, rather than a metric. The proposed loss function is differentiable and can be used in a Gradient descent framework. It was applied as part of a Neural Network, enabling non-linearity by having multiple layers.

The experiments prove the concept, for instance, they show that when a reduced training set is available, one benefits from using smaller values of lambda (the discrimination ratio), which is quite intuitive.

However, I felt that the experiments are a bit limited. It would have been interesting to see more practical applications of the proposed method, showing when it is relevant to be able to tune between robustness and discrimination power. Perhaps this method could be applied in domain adaptation datasets.

Minor style issues: - sentences starting with "And..." - missing spaces between words and brackets, e.g. "decreases(Fig. 3b)" - "attentions have" -> "attention has"

Note: I'm pleased to see that clarity has improved in the final version of the paper and that more details and evaluations are presented in the experimental results section.
Summary: The proposed loss function is interesting as it can be applied to any classification problem, particularly when it is necessary to control the trade-off between discrimination and robustness. This paper presents strong theoretical justification, but limited experimental validation.

Author Feedback
Author rebuttal: We thank all the reviewers for their positive and helpful comments. All issues are simple and will be addressed in the revision.

R1 concerns the novelty w.r.t. pure metric learning [7] which considers only discrimination without robustness. This is equivalent to setting lambda=1 in (3); it is a special case of our method. Experiments on LFW in Fig.5 and Tab.1 show that our method outperforms pure metric learning [7].

In the MNIST experiment, we exhibit the trade-off between discrimination and robustness when f_alpha is linear. The goal here is not superior performance to a nonlinear f_alpha, e.g., a deep neural network. The MNIST experiment is complementary to the synthetic data experiment, where f_alpha is nonlinear, showing that the discrimination-robustness trade-off is unavoidable even for linear f_alpha.

For the LFW experiment in Tab.1, we ran a new experiment using the deepface [14] cost function, and obtained an accuracy of 88.72%, inferior to our result 91.72%. We will provide details of the new experiment in the revised version.

R4 suggests experiments with more practical applications. We agree, and the LFW experiment addresses the issue of domain transfer. We have also added a new experiment using the deepface [14] cost function, obtaining an accuracy of 88.72%, inferior to our 91.72%. This shows the value of the discrimination-robustness tradeoff. The new experiment will be detailed in the revised version.

The minor issues will be fixed in the final version.

R7 suggests adding more comparison with state-of-the-art. We therefore ran a new experiment with a deepface [14] objective in the LFW experiment, obtaining an accuracy of 88.72%, inferior to our results 91.72%. It is noted that state-of-art results on LFW, e.g., deepface, are trained on huge proprietary dataset (we are using a different public training set). Therefore, it is not possible to repeat their network structure, which is adapted to their training set, and reproduce their results. For a fair comparison of the effectiveness of the objective functions, we adopt the same network structure as that of DML and cos-DRT.

We appreciate the very positive comments from R8.